# THE RIGHT LOSSES FOR THE RIGHT GAINS: IMPROVING THE SEMANTIC CONSISTENCY OF DEEP TEXT-TO-IMAGE GENERATION WITH DISTRIBUTION-SENSITIVE LOSSES

## ABSTRACT

One of the major challenges in training deep neural networks for text-to-image generation is the significant linguistic discrepancy between ground-truth captions of each image in most popular datasets. The large difference in the choice of words in such captions results in synthesizing images that are semantically dissimilar to each other and to their ground-truth counterparts. Moreover, existing models either fail to generate the fine-grained details of the image or require a huge number of parameters that renders them inefficient for text-to-image synthesis. To fill this gap in the literature, we propose using the contrastive learning approach with a novel combination of two loss functions: fake-to-fake loss to increase the semantic consistency between generated images of the same caption, and fake-to-real loss to reduce the gap between the distributions of real images and fake ones. We test this approach on two baseline models: SSAGAN and AttnGAN (with style blocks to enhance the fine-grained details of the images.) Results show that our approach improves the qualitative results on AttnGAN with style blocks on the CUB dataset. Additionally, on the challenging COCO dataset, our approach achieves competitive results against the state-of-the-art Lafite model, outperforms the FID scores of SSAGAN and DALL-E models by 44% and 66.83% respectively, yet with only around 1% of the model size and training data of the huge DALL-E model.

## 1 INTRODUCTION

The main aim behind the Text-to-Image generation (T2I) problem is to synthesize high-quality, photo-realistic images that semantically reflect input textual descriptions. It is a challenging computer vision problem that has many applications, including computer-aided design, image editing, and art generation. Most recent attempts at this problem utilize Generative Adversarial Networks (GANs) as the backbone model. Text-conditioned GANs have proven to be a powerful method to generate high-quality images that are semantically consistent with input captions. In practice, such models generate images that are significantly different from the ground truth. That's because, in most datasets, each image has several human-typed captions that are highly diverse in terms of content and structure. Also, models have to learn to understand two domains: textual description and visual description.

Most models in the literature either lack the details in the generated images or generate fine details in the image but with less accurate match to the textual description. The former problem happens due to employing a loss function that ensures sentence and word level matching between the image and the text such as the DAMSM loss (Xu et al., 2018; Ye et al., 2021; Zhu et al., 2019) without designing a generator capable of ensuring details are generated in the image at both fine and coarse grained levels. This leads to mainly washed images that have the general structure that matches the text (e.g., a bird with red wings) without the details that makes the image looks real such as the feathers and the color of the eyes. Moreover, these models mostly use multiple generators like Xu et al. (2018); Zhu et al. (2019); Qiao et al. (2019b) which can increase the aforementioned problem if the progressive growing of the details are not done to ensure that image features are learned well

at each step. This happens if the early generated image is poor, affecting the latter stages in the generator network.

Another problem with T2I is controllability and forcing the distribution of the generated images to be similar to the real ones. In general, we want similar textual descriptions to have similar image features and slight changes in the text to produce corresponding changes to the image without changing irrelevant features. Moreover, we want to push the generated images to look more like the real ones for all the given captions of the same real image. There are two approaches that can be done simultaneously to ensure these constrains: adding proper loss functions and changing the architecture of the generator like Hu et al. (2021).

This work aims to tackle these two problems by studying two directions: using a Style-based generator with the blocks of StyleGAN (Karras et al., 2020) either using the traditional style generator or fusing the style blocks with another architecture like AttnGAN to show how it will improve an already existing architecture. The aim of style blocks is to produce good fine and coarse grained features as well as give controllability of the generation as it did for its traditional use in the unconditional image generation. The other direction is to introduce contrastive learning in two flavors: real-to-fake contrastive learning and fake-to-fake contrastive learning. The purpose of these loss components is force the fake image distribution to be close to the real ones as well and to maximize the similarity between fake images for similar or the same caption. Experimentation with both directions lead to very promising results. Adding the style component evidently increase the quality of the generated images but produces low variability while adding the two flavors of contrastive losses gives a significant increase in visual quality of the generated images as well as the quality metrics (e.g., FID) when tried on the SSAGAN network (Hu et al., 2021), making it better than the reported state-of-the-art-models on the CUB birds dataset

## 2 RELATED WORK

### 2.1 TEXT-TO-IMAGE GENERATION

Great progress has been recently achieved in text-to-image generation by a large number of promising studies (Reed et al., 2016; Zhang et al., 2017; 2018a; Xu et al., 2018; Hong et al., 2018; Zhang et al., 2018b; Qiao et al., 2019b; Zhu et al., 2019; Yin et al., 2019; Li et al., 2019b; Tao et al., 2020; Qiao et al., 2019a; Li et al., 2019a; Cha et al., 2019; Hinz et al., 2019; El-Nouby et al., 2019; Liang et al., 2020; Cheng et al., 2020; Qiao et al., 2019b; Ramesh et al., 2021; Zhang et al., 2017; 2021), most of which employ GANs as the backbone model. In this section, we provide a summary of some of the most famous and relevant models. AttnGAN (Xu et al., 2018) utilizes attention to compute the similarity between the synthesized images and their corresponding captions using Deep Attentional Multimodal Similarity Model (DAMSM) loss. In this method, both the sentence and word-level information are used to compute the DAMSM loss. The stacked GAN architecture, proposed by Zhang et al. (2017), generates images incrementally from low-resolution to high-resolution. DM-GAN (Zhu et al., 2019) generates high-quality images using a dynamic memory GAN which refines the initial generated images. It then employs a memory writing gate to give more weight to relevant words and a response gate to enhance image representations accordingly. SD-GAN (Yin et al., 2019) uses a Siamese structure that takes a pair of captions as input and employs contrastive loss to train the model. For fine-grained image generation, SD-GAN adopts conditional batch normalization. Contrastive loss is also utilized in XMC-GAN (Zhang et al., 2021) but, unlike SD-GAN, they reduce training complexity by not requiring mining for information negatives. CP-GAN (Liang et al., 2019) adopts an object-aware image encoder together with a fine-grained discriminator for higher-quality image generation. While it achieves a promising Inception Score (Salimans et al., 2016), it has been shown to perform poorly when evaluated with a stronger FID (Heusel et al., 2017) metric. These approaches utilize several generators and discriminators to ultimately synthesize images at high resolutions. Other approaches have proposed inferring semantic layouts and explicitly generating different objects in hierarchical models (Hong et al., 2018; Hinz et al., 2019; Koh et al., 2021). In such models, generation is a multi-step process that requires more detailed labels (e.g., segmentation maps and bounding boxes), which represents a significant drawback. To tackle these issues, we employ a single generator and discriminator architecture in our model that is end-to-end trainable and generates much higher quality images.

## 2.2 STYLE BLOCKS

We discuss style-based image generation, focusing on the advances of the StyleGAN architecture. The StyleGAN (Karras et al., 2019) generator consists of two main components: the mapping and the synthesis networks. First, the mapping network transforms a noise vector Z into a latent space W; then, it applies affine transformations on W to generate styles A. This style code A is later used to enable scale-specific control of the synthesis process allowing for better disentanglement, enabling better control over the produced features. The synthesis network is the second major part of the StyleGAN generator, and it is responsible for generating the images. It consists of style blocks, where each block controls a specific level of details in the image based on style A. One of the regularization methods for style-based generators is style mixing. It works by feeding sampled styled codes A into different layers of the synthesis network independently at inference time to generate an image. The original StyleGAN (Karras et al., 2019) architecture used AdaIN to achieve style mixing; however, this resulted in artifacts in the generated images. In StyleGAN2 (Karras et al., 2020), this problem was tackled by basing the normalization on the expected statistics of the incoming feature maps. In other words, StyleGAN2 applies modulation and demodulation to the convolution weights within each style block, an alternative that removes artifacts while maintaining full controllability. Progressive Growing (Karras et al., 2017) was used for stabilizing high-resolution image generation, but it has its own characteristic artifacts and can impair shift equivariance. StyleGAN2 (Karras et al., 2020) tackled this problem by incorporating a multi-scale training technique. They tried different variations of the generator and discriminator to achieve that high-resolution synthesis, which was originally inspired by MSG-GAN (Karnewar & Wang, 2020) architecture. Based on an ablation the authors did, the winner network is using a skip generator and a residual discriminator.

## 2.3 TEXT-IMAGE CORRESPONDENCE

Part of the text-to-image generation problem is to maximize the semantic correspondence for image-text pairs by learning their joint representations. To that end, Deep Attentional Multimodal Similarity Model (DAMSM) (Xu et al., 2018) loss is used to learn such low-level text-image representations. The idea behind DAMSM loss is to train an image encoder and a text encoder simultaneously to encode certain words from the captions together with their corresponding sub-regions in the image to a common semantic space to ultimately compute the loss for image synthesis accordingly.

## 2.4 TEXT AND IMAGE ENCODERS

For text embeddings, Bi-directional Long Short-Term Memory (LSTM) (Schuster & Paliwal, 1997) has been recently utilized to extract semantic vectors from input captions. In Bi-directional LSTM, the semantic meaning of each word is represented by concatenating its two hidden states (one for each direction). Meanwhile, the global sentence vector is produced by concatenating the last hidden states from the Bi-directional LSTM. As for image encoding, most models in the literature use Convolutional Neural Networks (CNNs) to extract semantic vectors from images. The local image features of each image are learned by the intermediate layers of CNNs, while global features are learned from later layers. The Inception-v3 (Szegedy et al., 2016) model, pre-trained on ImageNet (Russakovsky et al., 2015), is commonly used as the image encoder, where local features are extracted from intermediate layers and global ones are extracted from the last average pooling layer. Ultimately, these image features are encoded together with text embeddings to a common feature space, where each image region is mapped to a visual feature vector.

## 2.5 CONTRASTIVE LEARNING

Contrastive learning has proven very effective in self-supervised representation learning of visual features through various contrastive methods (Arora et al., 2019; Chen et al., 2020b; He et al., 2020; Chen et al., 2020c; Khosla et al., 2020; Tian et al., 2020b; Robinson et al., 2020; Chuang et al., 2020; Hassani & Khasahmadi, 2020; Henaff, 2020; Kalantidis et al., 2020; Misra & Maaten, 2020; Tian et al., 2020a; Wang & Isola, 2020). Three major findings have been presented by SimCLR (Chen et al., 2020b) to learn better representations. First, introducing a composition of data augmentations. Second, adding a non-linear transformation (learnable) between the contrastive loss and the representation. Third, increasing the batch size and training step. Similar to Ye et al. (2021), we adopt this simple contrastive learning framework. We do so with a relatively small computational

cost and a simple implementation following the first proposal by SimCLR. We improve the quality of text representations by pushing together the captions that describe the same image and pushing away those that describe different ones.

# 3 METHODOLOGY

## 3.1 TEXT AND IMAGE ENCODERS

AttnGAN suggested a pair of encoders, one for the text and the other for image to be trained on the DAMSM loss in order to learn the suitable text and image encodings of the same space. The idea used here is by training a simple Bi-LSTM network along with a pre-trained sub-model from the Inception V3 model followed by a linear layer mapping the image encoding to the same dimensions of the text encoder. Using this combination allows the Bi-LSTM to learn suitable mapping from the pure tokenized text to the image encoding space representation. We tried improving on this solution by replacing the LSTM with BERT encoding. However, the models failed to converge since the layer after the image encoder was not enough to map the image encodings to the BERT encodings. Therefore, for the rest of our paper, we decided to use the pre-trained Bi-LSTM provided by Ye et al. (2021).

## 3.2 ATTNGAN STYLE-BASED ARCHITECTURE

Due to the state-of-the-art results achieved using StyleGAN in style-based image generation, we decided to use a combination of the style blocks with the attention mechanism of AttnGAN. The modified architecture can is shown in Figure 1. The combination is done by adding a mapping network composed of eight fully connected layers to map the text encoding and the random noise to a disentangled space W. The rest of the generator architecture is based on replacing the normal generator block from AttnGAN with StyleGAN2 blocks. Similarly, the discriminator blocks of AttnGAN are replaced with the Style based discriminator blocks.

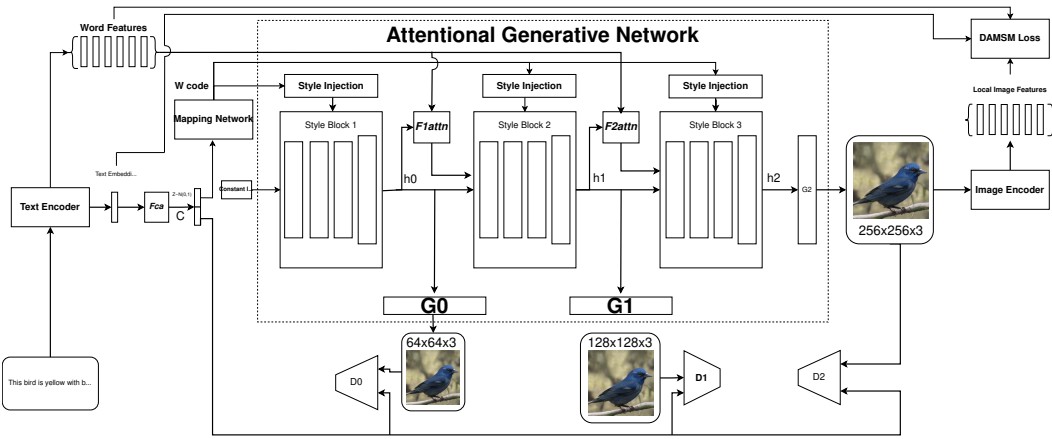

Figure 1: AttnGAN with Style Blocks

## 3.3 CONTRASTIVE LOSS ON SSAGAN ARCHITECTURE

SSA-GAN (Hu et al., 2021) follows the one generator-discriminator pair architecture to tackle the issue of image quality in stacked generator-discriminator architectures. The core element of this framework is the SSA-CN block, which uses Semantic-Spatial Condition Batch Normalization and a mask predictor to fuse the text and image features effectively and deeply.

Each SSA-CN block consists of an up-sample block, a residual block, a mask predictor, and a Semantic-Spatial Condition Batch Normalization (SSCBN) block. It predicts a mask map that indicates which parts of the current image feature maps still need to be reinforced with text information so that the refined image feature maps are more semantically consistent with the given text. The

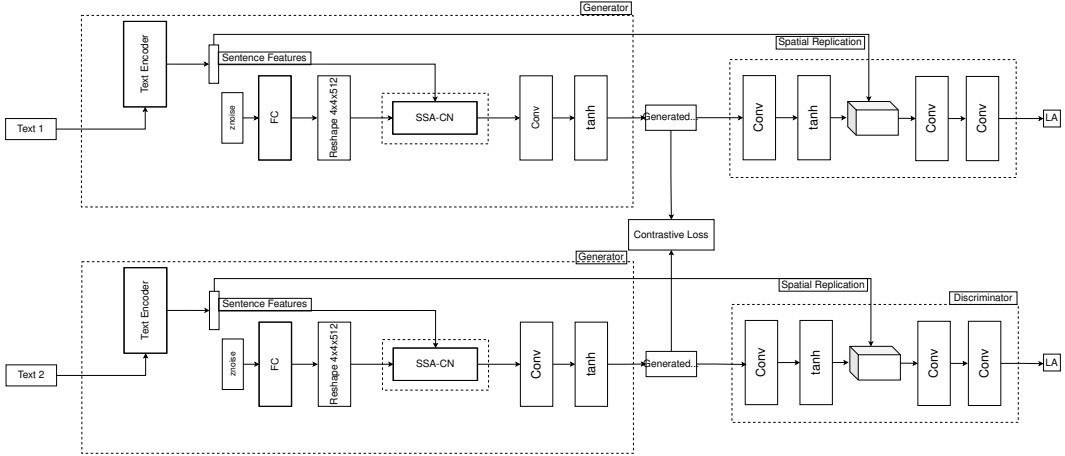

Figure 2: SSAGAN + Contrastive Loss

SSCBN achieves semantic and spatial conditional batch normalization, which leads to accurate and deep text and image features fusion as needed.

In this paper, we introduce two contrastive losses to the current SSA-GAN (Hu et al., 2021) architecture (as shown in Figure 2). We define the fake-to-fake contrastive loss between a pair of generated images using semantically similar captions. This allows us to increase the semantic consistency between images generated from related descriptions, while maximizing the distance between those generated from different descriptions.

Similarly, we are calculating the real-to-fake contrastive loss between the generated images and the real ones, aiming at reducing the semantic difference between our generated images and the ground truth. (as shown in Figure 3).

Combining these contrastive learning methods, we are able to generate semantically consistent images when given semantically similar captions; in addition, we also decreased the discrepancy between the distributions of the real and fake images.

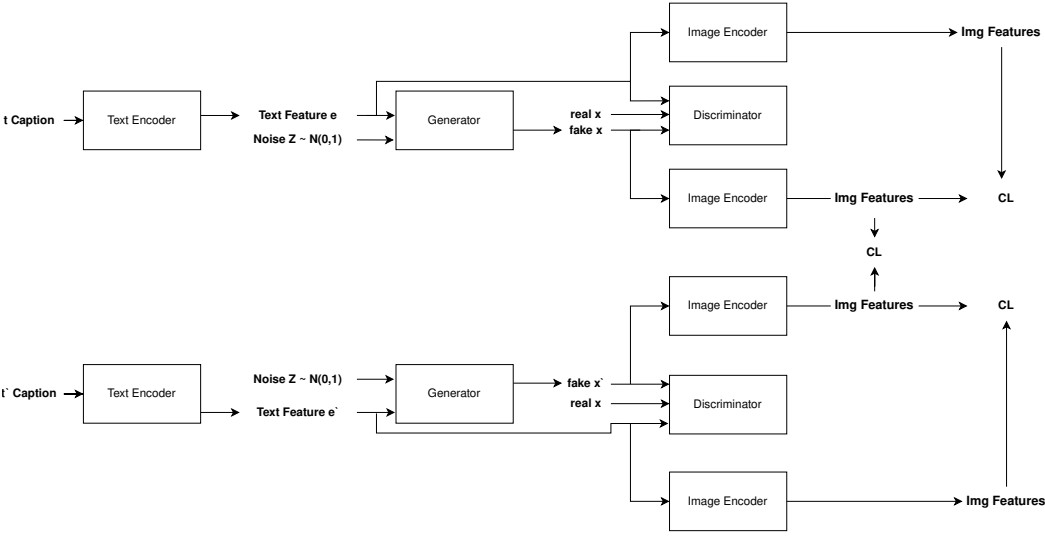

Figure 3: Contrastive Loss Architecture

### 3.4 LOSS FUNCTIONS

We use the normal adversarial loss for both the generator and the discriminator networks. Any of the other loss functions are added on top of the generator loss. We also use the DAMSM Loss defined in (Xu et al., 2018) as is with only changes to the weight applied to it. Next, we define three other loss functions.

#### 3.4.1 FAKE-TO-REAL CONTRASTIVE LOSS

The fake-to-real contrastive loss is introduced to minimize the distance between the encoding of the ground truth image and the encodings of the synthesized images to ensure images similar to the real image are being generated and vice versa.

We utilize the Normalized Temperature-scaled Cross-Entropy Loss (NT-Xent) (Sohn, 2016; Wu et al., 2018; Chen et al., 2020a) as the contrastive loss. For a pair of fake and real images $a$ and $b$, let $sim(a, b) = a^T b / \|a\| \|b\|$ denote the dot product between $l_2$ normalized $a$ and $b$. The loss function for the $i$th sample is as follows:

$$L(i) = -\log \frac{exp(sim(u_i, u_j)/\tau)}{\Sigma_{k=1}^{2N} 1_{k \neq i} exp(sim(u_i, u_k)/\tau)} \tag{1}$$

where the $i$th and $j$th samples make the positive pair, $1_{k \neq i}$ is an indicator function whose value is 1 if and only if $k \neq i$, $\tau$ is a temperature parameter, and N is the batch size. The overall contrastive loss is computed across all positive pairs in a mini-batch, which is computed as follow:

$$L_C = \frac{1}{2N} \Sigma_{i=1}^{2N} L(i). \tag{2}$$

#### 3.4.2 FAKE-TO-FAKE CONTRASTIVE LOSS

The fake-to-fake contrastive loss is adopted to minimize the distance between the encodings of generated images that are conditioned on two captions related to the same image and maximize those conditioned on captions related to different images. This approach ensures that the produced images are more similar to each other for the input captions related to the same image. In a very similar fashion, We use the NT-Xent loss introduced earlier in Equation 1 in order to calculate the overall fake-to-fake contrastive loss.

#### 3.4.3 IMAGE RE-CAPTIONING LOSS

The previous two losses aim to have the model converge on having similar features between the images of the same set of captions and between the real and the fake images. The third loss aims to verify that the image, when fed to a captioning model, will have the same caption features as the original model. Let $F_C$ be the captioning model, the loss can be written as:

$$L_{CP} = L_e(F_C(Fake_i), C_i) \tag{3}$$

where $Fake_i$ is the fake image, $C_i$ is the correct caption, and $L_e$ is the normal Cross Entropy Loss.

#### 3.4.4 OVERALL LOSS FUNCTION

After the pretraining of the image encoder and the text encoder using the DAMSM + Contrastive Loss functions, the image encoder and text encoder spaces should be able to produce similar representations (image and text are projected to a similar space). Let $L_G$ be the generator loss, $L_{DAMSM}$ be the DAMSM loss, $L_{CP}$ be the recaptioing loss, $L_{CR}$ be the contrastive real-to-fake loss (F2R), $L_{CF}$ be the contrastive fake-to-fake loss (F2F) illustrated in figure 3. Then, the overall loss function is:

$$L = L_G + \lambda_1 \cdot L_{DAMSM} + \lambda_2 \cdot L_{CR} + \lambda_3 \cdot L_{CF} + \lambda_4 \cdot L_{CP} \tag{4}$$

Where $\lambda_1 \ldots \lambda_4$ are the weights of the losses. We can turn of the losses we don't want by setting their weight to zero so that we can test the effect of individual loss components. Our typical values for these weights are $\lambda_2 = \lambda_3 = 0.2$, $\lambda_4 = 1$, $\lambda_1$ is about 0.05 for the SSA-GAN architecture and 5 for the Style based architecture.

## 4 EXPERIMENTAL SETUP

### 4.1 DATASETS

**The Caltech-UCSD Birds 200** (CUB-200, or CUB for short) (Wah et al., 2011) is an image dataset of 200 bird species. The CUB dataset has almost 13k images of birds belonging to 200, mostly North American, bird species and a textual description for each image. CUB has one class of images, which made it easier to train on and provided a simpler mapping space for the generator network.

**The Microsoft Common Objects in Context** (MS COCO, or COCO for short) dataset (Lin et al., 2014) is a large-scale object detection, segmentation, keypoint detection, and captioning dataset. The dataset consists of 328K images of 80 object categories. The numerous COCO object classes make it more challenging than the CUB-200 dataset and more difficult for text-to-image models to converge on. Additionally, the hardware requirements and the time needed to train a model on the COCO dataset are substantially more than those required by the CUB dataset, to an extent that was not available to us. Such limitations urged us to initially train our different model architectures on the CUB dataset before training them on the COCO dataset. Accordingly, some models were not trained on the COCO dataset; yet AttnGAN and SSAGAN, both with and without the contrastive loss, and SSAGAN with the real-to-fake contrastive loss were trained.

### 4.2 METRICS

Following previous work, we now introduce the Frechet Inception Distance (FID), the Inception Score (IS), and the R-precision metrics we adopted to quantitatively evaluate the performance of our work.

#### 4.2.1 QUALITY METRICS

**FID** (Heusel et al., 2017) is a prevalent evaluation metric used to evaluate the quality of generated images from GANs. Specifically, it measures the Frechet distance between the distribution of the synthetic images and the real images in the feature space of a pre-trained Inception v3 network (Heusel et al., 2017). The Frechet distance, also known as the Wasserstein-2 distance when the two distributions are normal distributions (Kynkäänniemi et al., 2022), is a distance function used to compare the probability distributions of two variables or to measure the similarity between curves taking into account the location and ordering of points along the curves. The Frechet distance between two multivariate Gaussian distributions $X$ and $Y$ is:

$$FID = \|\mu_x - \mu_y\|^2 - Tr(\Sigma_x + \Sigma_y - 2\Sigma_x\Sigma_y), \tag{5}$$

, where the means and covariance matrices of $X$ and $Y$ are $mu_x$, $mu_y$ and $\Sigma_x$, $\Sigma_y$, respectively, and $Tr$ is the trace of the matrix.

**IS** (Salimans et al., 2016) is another ad-hoc evaluation metric of the quality of image generative models. The IS uses a pre-trained Inception v3 network and calculates the conditional probabilities for each generated image p(x—y). To that end, the Kullback-Leibler (KL)-divergence is computed for each image between the conditional and marginal class distributions:

$$KL - divergence = p(y|x) * (log(p(y|x)) - log(p(y))) \tag{6}$$

, where p(y—x) is the conditional class distribution, and $p(y) = \Sigma_x p(y|x)p_g x$ is the marginal class distribution, where $p_g x$ is the distribution of generated images. That is, the IS score is computed by:

$$IS = exp(E_{x \sim p_g} D_{KL}(p(y|x)||p(y))). \tag{7}$$

| Architecture | IS ↑ | FID ↓ | R-Precision↑ |
|---|---|---|---|
| AttnGAN | 4.36 | 23.98 | 58.06 |
| AttnGAN + CL | 4.42 | 16.34 | 60.52 |
| Style1AttnGAN | 4.20 | 36.82 | 61.25 |
| Style1AttnGAN + CL (F2F) | 3.09 | 41.32 | 62.30 |
| Style2AttnGAN + CL (R2F) | 3.59 | 38.51 | 64.53 |
| SSAGAN | 4.89 | 9.76 | 75.67 |
| SSAGAN + CL (F2F) | 4.95 | 9.62 | 70.49 |
| SSAGAN + CL (R2F+F2F) | **5.18** | **9.12** | **72.85** |
| SSAGAN + CL (R2F+F2F+R) | 5.03 | 9.16 | 72.28 |

Table 1: Comparison between the performance of different models when trained on the CUB dataset with different combinations of loss functions.

| Architecture | IS ↑ | *FID* ↓ | *R-Precision* ↑ |
|---|---|---|---|
| AttnGAN | 6.08 | 30.67 | 83.8 |
| AttnGAN + CL | 6.29 | 26.89 | 84.24 |
| SSAGAN | 7.18 | 19.37 | 82.45 |
| SSAGAN + CL (R2F) | 7.6 | 12.08 | 85.20 |
| SSAGAN + CL (R2F + F2F) | **7.62** | **10.89** | **88.21** |

Table 2: Comparison between the performance of different models when trained on the COCO dataset with different combinations of loss functions.

### 4.2.2 ACCURACY METRICS

**R-precision**    (Xu et al., 2018) is a standard evaluation measure for ranking retrieval results of a system. It is defined as the top-R retrieved documents that are relevant, where R is the number of relevant documents for the current query. In other words, R-precision is r/R when there are r relevant documents among the top-R retrieved documents. Similarly, R-precision is used in text-to-image generation tasks to measure the correlation between a generated image and its corresponding text. R-precision is especially important since the FID and the IS cannot reflect the relevance between an image and a text description. Specifically, we use the generated images to query their corresponding text descriptions. The image encoder and text encoder learned in the pre-trained DAMSM are then utilized to extract global feature vectors of the generated images and their text descriptions, which are used to compare the Cosine similarities between them. Finally, candidate text descriptions for each image are ranked in descending order of similarity, and the top r relevant descriptions are found to compute the R-precision. Each model generates 30,000 images from randomly selected unseen text descriptions to compute the IS and the R-precision. The candidate text descriptions for each query image consist of one ground truth (i.e., R = 1) and 99 randomly selected mismatching descriptions.

## 5    RESULTS AND DISCUSSION

We first tackled the T2I generation problem on the CUB dataset because it contains birds only, which is easier to tackle than the multi-class COCO dataset. We compare our proposed model against several architectures; however, we will focus on the comparison with SSA-GAN because it is the baseline model. The main quantitative results can be shown in Table 1, while the qualitative results can be shown in Figure 3. The variants of our proposed model notably outperform the SSA-GAN model on both FID and IS metrics. Table 1 also shows that adding the Real-to-Fake contrastive loss leads to lower FID and higher IS and R-precision compared to using only Fake-to-Fake contrastive loss. It is also worth noting that the qualitative results 3 of our approach are better than the baseline results. The generated images are more diverse, semantically closer to the captions, and contain fine-grained and coarse details. Table 2 shows our quantitative results on the more challenging COCO dataset, while Figure 4 shows the qualitative results. The results on the COCO dataset also show that our model significantly outperforms the baseline model. Particularly, the FID score is 11% better than the SSAGAN+CL (R2F) model on the COCO dataset. Comparing this to the 5% enhancement on the CUB dataset, we can conclude that our approach performs even better when the

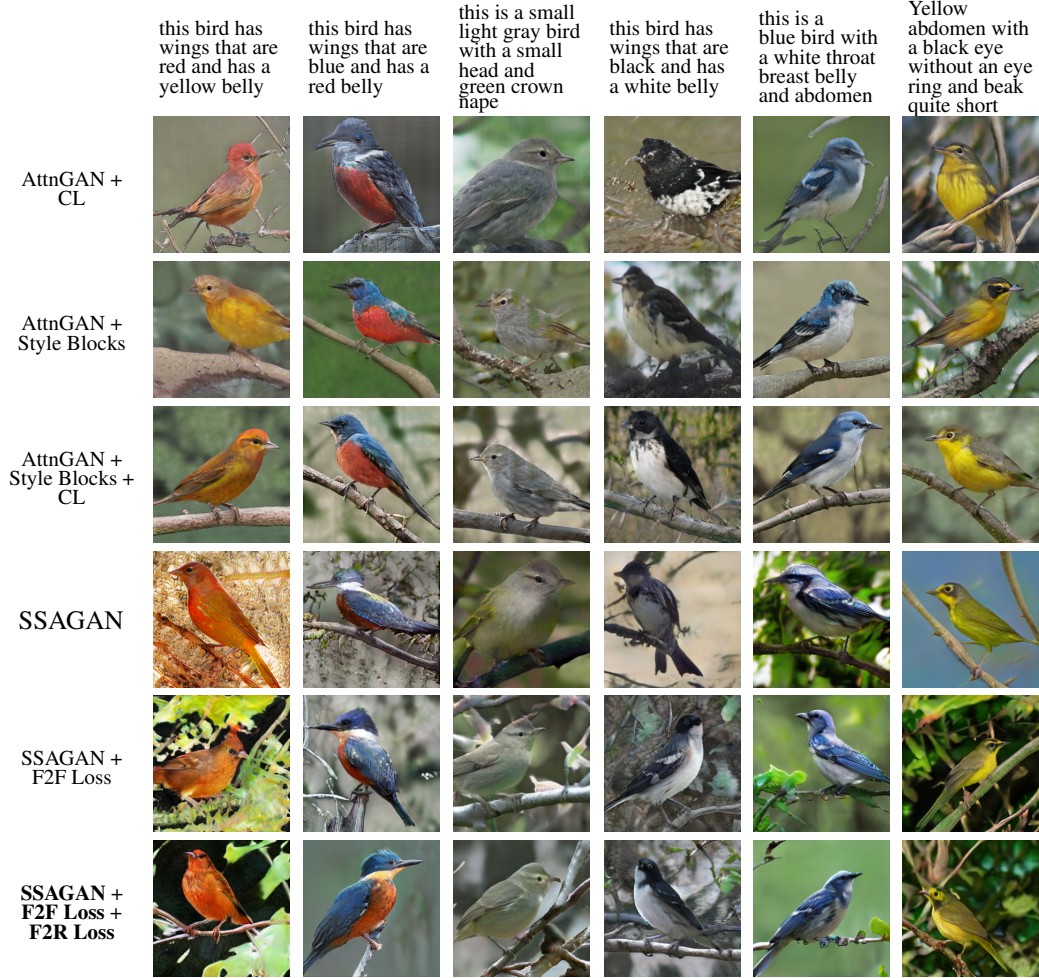

Table 3: Comparison of example images between our approach and baselines on the CUB dataset

problem gets harder. We emphasize that, on the challenging COCO dataset, our approach achieves competitive results against the state-of-the-art Lafite model, outperforms the FID scores of SSA-GAN and DALL-E models by 44% and 66.83% respectively, yet with only around 1% of the model size and training data of the huge DALL-E model.

## 5.1 WHY STYLEGAN SCORES ARE LOW

As seen consistently in the results tables 1 and 2, despite that adding style blocks to AttnGAN architecture has a significant impact on the quality and level of details of generated images, the quality metrics (FID and IS) are low compared to AttnGAN. After several experiments, we can see that this is because the variance of generated images is low compared to the other architectures. While the network focuses on learning to represent both fine and coarse grained details, it learns a small number of poses, sizes and shapes of birds and a limited set of backgrounds. This leads to a notable decrease in the variance and consequently larger FID values. We attribute this behavior to the fact that disentanglement of the noise via the mapping network as well as the style injection make the generator focus on the visual features specified in the text like colors and sizes and ignore those that are not in the text like poses and backgrounds. It is important to emphasize that the image-text matching was learned well by the network, so the only issue was the variability of the outputs. Moreover, the effect of contrastive learning was very evident on the visual quality in the style based architecture.

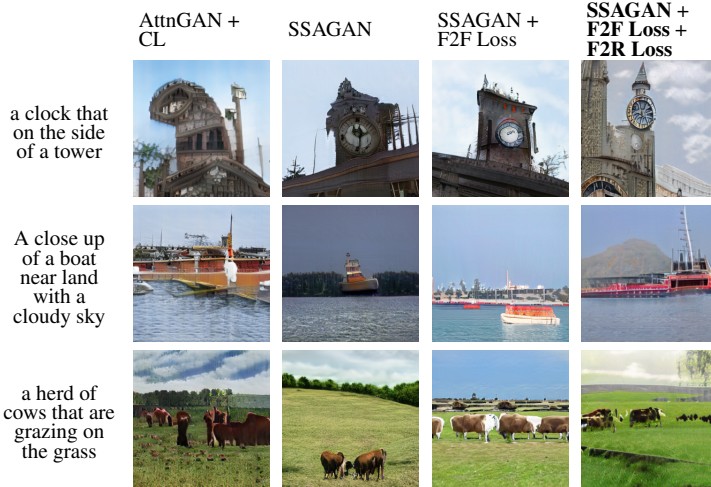

Table 4: Comparison of example images between our approach and baselines on the COCO dataset.

## 6 CONCLUSION

In this paper, we propose a novel approach for text-to-image generation using contrastive learning. Our goal is to address the semantic inconsistency between generated images and their corresponding captions while keeping the fine-grained details and maintaining a small model size. We achieve this goal by incorporating a mix of two distribution-sensitive contrastive losses: fake-to-fake and fake-to-real losses. The fake-to-fake loss increases the invariability of generated images to linguistic changes and the fake-to-real loss increases image quality and relevance to the text. To test our hypothesis, we use two generative baselines: SSA-GAN and a modified version of AttnGAN in which style blocks are incorporated. Our results show an increase in the quality of generated images, surpassing the SOTA FID results using both contrastive learning and style mixing on the CUB dataset. In addition, we achieve competitive results on the COCO dataset, outperforming the FID score of our baseline SSAGAN model by 44%, and that of DALL-E by 66.83% a with much smaller model capacity and training data.

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
