# OpenReview forum: "The Right Losses for the Right Gains: Improving the Semantic Consistency of Deep Text-to-Image Generation with Distribution-Sensitive Losses"
_ICLR.cc/2023/Conference — Submitted to ICLR 2023_

### Official Review · Reviewer_Bi56 · 2022-10-21

**Confidence:** 4
**Correctness:** 3
**Technical Novelty And Significance:** 2
**Empirical Novelty And Significance:** 2
**Recommendation:** 3

**Clarity, Quality, Novelty And Reproducibility:**

Overall, this paper is well-written and it is easy to follow. However, the motivation is not convincing and the experiments are not strong enough to support their claim. Please find more details in Strength And Weaknesses.

**Strength And Weaknesses:**

### Strength
This paper is overall well-written and it is easy to follow.

### Weaknesses

**Novelty**

1. The core idea of this paper is the use of StyleGAN block and contrastive learning. However, these two techniques are not new for image synthesis. The novelty of the proposal is limited.

**Clarity**

1. The motivation is not clear. The authors mentioned in the second paragraph on page 1 that, the use of DAMSM loss leads to washed images without details. However, DAMSM loss is usually used to match text and images, it is not clear why it will hurt details.
2. The motivation for using fake-to-fake contrastive learning is not clear. For example, given the text ‘dog’, there may be many different kinds of dogs that correspond to this text. Users may also need diverse results given the same caption. Forcing similar results for the same caption may lead to mode collapse.
3. The motivation for using real-to-fake contrastive learning is not clear. As claimed on page 2, the authors use real-to-fake contrastive learning to push fake distribution close to the real distribution. However, the generative adversarial loss has worked for this motivation.

**Quality**

1. The authors claimed the benefit of generation controllability on Page 2, however, they forget to verify the generation controllability in experiments.
2. The authors ignore comparisons with recent text-to-image works, e.g., DALLE, and some other diffusion-based text-to-image works, e.g., Imagen, etc.
3. The results are not strong enough to verify the effectiveness of the proposed approach. For example, in the first case of Figure 3, the proposed approach shows the heavy color bleed issues and generates a red belly for the ‘yellow belly’ text input.


**Summary Of The Paper:**

This paper aims to generate fine-grained details and semantically match images with the text input. Specifically, the authors propose using a StyleGAN-like block with existing generators to improve details synthesis. Besides, the authors propose fake-to-fake and real-to-fake contrastive loss to improve semantic correspondences. They conduct both quantitative and qualitative experiments on CUB and coco.


**Summary Of The Review:**

This paper aims to synthesize more fine-grained details and semantically match images for the input text by StyleGAN-based modules and contrastive learning. However, the motivation is not convincing and the experiment results are not strong enough. If the authors address all the weaknesses mentioned in Strength And Weaknesses, I will consider raising my rating

---

### Official Review · Reviewer_f3ju · 2022-10-25

**Confidence:** 5
**Correctness:** 2
**Technical Novelty And Significance:** 2
**Empirical Novelty And Significance:** 2
**Recommendation:** 3

**Clarity, Quality, Novelty And Reproducibility:**

Novelty of proposed method is limited, and the paper lacks sufficient experimental results to support authors' statements.

**Strength And Weaknesses:**

Strength:
1. Authors show the effectiveness of both style-based technique and contrastive learning used in multi-modal image generation task.

2. Sufficient related works are discussed in the paper.

Weaknesses:

1. The baselines adopted in the paper might not be state-of-the-art, for example, authors do not include XMC-GAN, GLIDE, etc. Also, authors claim that "their method achieves competitive results against the state-of-the-art Lafite model, outperforms the FID scores of SSA- GAN and DALL-E models by 44% and 66.83% respectively, yet with only around 1% of the model size and training data of the huge DALL-E model", but there are no corresponding experimental results shown in the paper.

2. The qualitative results on COCO shown in the paper might not be good enough.

3. The novelty of proposed style block is limited, which mainly depends on StyleGAN. Also the paper lacks a sufficient description of the style block. Contrastive losses have been studied in XMC-GAN. Authors might discusses the difference between their method and XMC-GAN

4. Could authors give more details about why the proposed method can greatly improve the performance?




**Summary Of The Paper:**

The paper shows that the adoption of contrastive learning can improve the text-image semantic consistency and the quality of synthetic images, and the adoption of style block can enhance the fine-grained details of the images. The experiments by implementing both methods on AttnGAN and SSAGAN shows the effectiveness of them.

**Summary Of The Review:**

See above weaknesses. I am happy to change my rating based on authors' responses.

---

### Official Review · Reviewer_m8jC · 2022-11-01

**Confidence:** 4
**Correctness:** 4
**Technical Novelty And Significance:** 1
**Empirical Novelty And Significance:** 1
**Recommendation:** 3

**Clarity, Quality, Novelty And Reproducibility:**

The writing can be improved.
The novelty is limited.

**Strength And Weaknesses:**

Strengths:
1. This paper proposes to use contrastive learning to improve the semantic consistency of text-to-image synthesis models.
2. It achieves good results on CUB and COCO datasets.

Weaknesses:
1. The writing can be improved. The current version is not polished and difficult to follow.
2. The contribution is limited. Contrastive learning has been explored extensively in previous literature, and only applying the contrastive learning to text-to-image synthesis pipelines is not enough novelty and contribution.
3. The visual quality of synthesized images is limited.

**Summary Of The Paper:**

This paper proposes to use contrastive learning to improve the semantic consistency of text-to-image synthesis models. In the proposed approach, the fake-to-fake loss is adopted to increase the semantic consistency between generated images of the same caption, and fake-to-real loss is adopted to reduce the gap between the distributions of real images and fake ones. The approach is built upon SSAGAN and AttnGAN, and experiments are conducted on CUB and COCO datasets.

**Summary Of The Review:**

The paper simply combines contrastive learning with text-to-image synthesis frameworks, and it is of limited novelty and contribution.

---

### Official Review · Reviewer_FV8R · 2022-11-02

**Confidence:** 3
**Correctness:** 3
**Technical Novelty And Significance:** 2
**Empirical Novelty And Significance:** 2
**Recommendation:** 5

**Clarity, Quality, Novelty And Reproducibility:**

- Clarity is decent.
- Quality-wise, please refer to the "weakness" section.
- Novelty: my concern is that the proposed method is a combination of other readily-available methods. But the plus side is that the authors often provide justification of why certain methods are adopted.

**Strength And Weaknesses:**

Strength:
- Overall easy to follow
- This work shows great empirical results, and the proposed method is compared against multiple metrics
- Thorough literature review
Weakness:
- While the design of the loss function intuitively makes sense, I have concerns about the weights assigned to those losses: 1) the assignment seems quite heuristic, and it would be good if the authors provide more justification on why certain losses should be given a higher weight; 2) the weight of DAMSM is significantly higher than other weights when style-based models are used - this choice will "dilute" the other loss functions emphasized by the authors, and I wonder why the authors make this decision
- It would be nice if style based models are also compared with in the COCO task
- DALL-E's performance should also be presented in Table 1 and 2
- In Table 1, SSAGAN actually has a better score in R-precision

**Summary Of The Paper:**

This work proposes a text-to-image generation model that leverages several distribution-sensitive losses for better performance under smaller model sizes. The major technical contribution is the design of the "right" loss function for this task, which is a linear combination of generative loss, DAMSM, fake-to-real, fake-to-fake, and re-caption loss. The authors propose that the model significantly outperforms DALL-E in terms of FID score but only has 1% of its model size and training data.

**Summary Of The Review:**

Overall it's a clear paper. The technical novelty is not the strongest, and the empirical results could be demonstrated in a more convincing way. More rationales on the designs of the loss function, and more thorough comparison with other methods will put this work in a much stronger stance.

---

### Decision · Program_Chairs · 2023-01-20

**Decision:**

Reject

**Justification For Why Not Higher Score:**

All the reviewers recommend rejection of the paper, and no author rebuttal is provided. There is no reason to accept the paper.

**Justification For Why Not Lower Score:**

N/A

**Metareview: Summary, Strengths And Weaknesses:**

This paper proposes the use of two contrastive learning losses to improve the text-image semantic consistency and the quality of synthetic images, and the adoption of style block to enhance the fine-grained details of the images.

It received scores of 3335. All the reviewers are negative about the paper. Some major concerns include: (1) limited novelty of using contrastive losses, (2) the motivation is not clear, (3) results are not strong, and (4) the baselines adopted are not SoTA methods.

No rebuttal is provided, therefore, there is no reason to accept the paper. The AC would like to recommend rejection of the paper.